# Latent Learning Progress Drives Autonomous Goal Selection in Human Reinforcement Learning

**Gaia Molinaro**
University of California, Berkeley
gaiamolinaro@berkeley.edu

**Cédric Colas**
Massachusetts Institute of Technology
ccolas@mit.edu

**Pierre-Yves Oudeyer**
Inria Centre at the University of Bordeaux
pierre-yves.oudeyer@inria.fr

**Anne G. E. Collins**
University of California, Berkeley
annecollins@berkeley.edu

## Abstract

Humans are autotelic agents who learn by setting and pursuing their own goals. However, the precise mechanisms guiding human goal selection remain unclear. Learning progress, typically measured as the observed change in performance, can provide a valuable signal for goal selection in both humans and artificial agents. We hypothesize that human choices of goals may also be driven by *latent learning progress*, which humans can estimate through knowledge of their actions and the environment – even without experiencing immediate changes in performance. To test this hypothesis, we designed a hierarchical reinforcement learning task in which human participants (N = 175) repeatedly chose their own goals and learned goal-conditioned policies. Our behavioral and computational modeling results confirm the influence of latent learning progress on goal selection and uncover inter-individual differences, partially mediated by recognition of the environment's hierarchical structure. By investigating the role of latent learning progress in human goal selection, we pave the way for more effective and personalized learning experiences as well as the advancement of more human-like autotelic machines.

## 1 Introduction

Animals often spontaneously explore their environment in the absence of immediate extrinsic payoffs [1–8]. While seemingly wasteful, such intrinsically motivated behavior is argued to adaptively guide learning and exploration, paying off across developmental and evolutionary timescales [9–11]. As a result, researchers and engineers started endowing artificial systems with intrinsic reward signals that aim to reproduce aspects of human curiosity-driven learning [11–17]. A striking aspect of human intrinsic motivation is *autotelic exploration*, wherein people self-generate goals, and reward themselves for achieving them [18–20]. How people decide which goals to pursue is thus an outstanding issue in the cognitive sciences [21]. Identifying the key variables affecting goal selection may in turn enable the development of artificial agents that intelligently set their own objectives, rather than being optimized for predefined ones [22].

Alongside factors such as performance and novelty, learning progress (LP) – the derivative of performance with respect to time – has proven a useful signal for goal selection, steering both humans [23, 24] and artificial agents [25–28] away from tasks that are either too simple or too hard. Most existing approaches approximate LP using past observations (e.g., differences in recent and earlier performance [23]), such that LP is greatest when performance changes rapidly ([29], but see [30]). However, there are often situations where no external change is visible, yet some other form of progress toward the desired outcome is made. Imagine being tasked with identifying the correct

38th Conference on Neural Information Processing Systems (NeurIPS 2024).

sequence of numbers to open a combination lock, which you might attempt through trial and error. Throughout most of this scenario, repeated failures would yield no difference in performance, hence no empirical learning progress (as typically defined). Nonetheless, provided that the lock has a limited number of slots and numbers and that you can avoid repeating incorrect combinations, every attempt is a step toward the solution. Based on this observation, we propose extending the definition of LP to include a *latent* variant, distinct from its manifest counterpart in that it relies on an internal model of the environment and one's behavior in addition to performance. Following this example, we formulate latent learning progress (LLP) as a measure of progress an agent infers by leveraging knowledge about the environment and its interactions with it, e.g., through reasoning. For example, LLP could be tracking the proportion of hypotheses tested over the entire solution space, thus changing with every new set of actions taken and dropping upon identifying the correct one. Unlike LP, LLP does not require observing performance improvements to provide informative signals about progress. At least in certain settings, LLP could be a more valuable signal for goal selection than the classically defined LP, as it proactively estimates future progress without waiting for performance improvements.

Here, we hypothesize that LLP is a signal humans use for goal selection, and that it can be considered separately from LP. Specifically, we test the role of LLP in human goal selection by introducing a novel learning paradigm with goal choices as the key dependent variable. Several features of this environment (detailed in Section 3.1) make it possible to distinguish between LP and LLP and test the latter's contributions to autotelic exploration in a sample of human participants. Among these, a hierarchical component is introduced in the learning environment to further target LLP, which might be sensitive to hierarchy-based inferences leading to sharp changes in the agents' knowledge of the environment. These features make our environment suited for probing LLP and testing its contributions to autotelic exploration in a sample of human participants. By combining behavioral findings and computational modeling of human decisions in the experiment, we find support for our hypothesis that LLP captures human goal choices better than standard LP in this hierarchical reinforcement learning task. We also find inter-individual differences in goal selection strategies, warranting further studies on the factors underlying such idiosyncrasies. In our discussion of the results, we argue that adopting the notion of LLP may foster the development of autotelic machines.

## 2   Related work

**Intrinsic motivation**   The various forms of intrinsic motivation can broadly be classified as either knowledge-based (KB) or competence-based (CB) [12, 28, 29, 31, 32]. KB intrinsic motivations derive from comparisons between pre-existing and newly acquired information and are well-documented in the animal world (e.g., [1, 3, 5, 6, 8, 33–37]). In artificial agents, KB signals of novelty, diversity, or prediction error broaden exploration when provided alongside extrinsic rewards [13, 15, 16, 38, 39] or even on their own ([11, 14], see [40] for review). However, human curiosity is often driven towards stimuli of intermediate complexity [7, 41, 42], rather than extremes. Accordingly, some artificial models aim to maximize an "intermediate" difference between predictions and observations [30, 43]. Intrinsic motivations based on learning progress (LP) can avoid arbitrary cutoffs for what signifies intermediate complexity. LP is particularly useful as a CB motivation, prompting agents to improve their performance on self-determined goals [29, 32, 44]. Indeed, humans often find pleasure in activities whose difficulty matches their abilities [45, 46]. CB intrinsic motivations highlight the autotelic property of human learning, which is often driven by self-defined objectives. Two concurrent challenges in cognitive science and artificial intelligence involve identifying the signals people use to develop their learning curricula [19, 21] and adapting those principles to machines [22].

**Goal selection in humans**   Goals, defined as "a cognitive representation of a future object that the organism is committed to approach" [47], have long been a key concept in psychology (e.g., [48–51]), with a rich literature on how various types of goals interact with performance and motivation [52, 53]. In cognitive neuroscience, goals are typically imparted by the experimenter and are not the main focus of the analyses [54] – though recent work illustrated the impact of goals on core aspects of neural, cognitive, and behavioral responses ([18, 20, 21, 55–59]; see [19] for review). Nevertheless, among the principles thought to guide human goal selection is the tendency to choose activities that balance the expected outcome's desirability and the probability of attaining it [60]. When free to allocate their time to various tasks or choosing activities for pure enjoyment, people naturally tend to match the activity's level of challenge to their current abilities [45, 46, 61], which might be optimal for learning [62, 63]. However, most studies fail to provide precise computational details on the mechanisms of

goal selection. Although not explicitly addressing goals, two recent experiments attempted to fill this gap. In particular, [23] tasked participants with repeatedly choosing which of four learning activities (each with a different difficulty level) to engage with, and found that performance and LP jointly contribute to activity selection. Similarly, [24] found that people use novelty and LP when exploring various activities based on their own curiosity. Here, we propose that in deciding which goals to set, humans may rely not only on manifest signals of performance but also on an internally tracked, latent form of learning progress.

**Goal selection in artificial agents**    Enabling agents to learn autonomously has been a longstanding challenge in artificial intelligence and a central tenet of many successes in the field [43, 64–68]. However, most existing models are optimized for a predetermined objective. Overcoming this limitation, autotelic agents are designed to autonomously select goals and learn to achieve them [22, 25, 69], begging the question of how to optimally set goals for efficient learning. Even random goal sampling [70–72] or entropy-maximization objectives [73, 74] present some advantages [75], but agents can be programmed to adjust their goal selection to currently owned skills and further benefit from autotelic properties. One such approach involves a two-part adversarial system, wherein a "teacher" proposes challenging tasks and a "student" learns to maximize performance over them [44, 76–78]. Algorithmic implementations of LP also help agents avoid wasting time and resources on objectives that are too far-reaching or even impossible (e.g., due to inherent noise in the system; [26–28, 79–85]). These approaches calculate LP relative to performance, not considering changes in the agent's goal-conditioned competence that might occur latently. Thus, to our knowledge, LLP-mediated goal selection in artificial agents is an unexplored avenue.

## 3 Experiment design and modeling approach

### 3.1 Experimental setup

Most reinforcement learning problems consider a single task to master. Similarly, experimental psychologists usually instruct participants to pursue a specific objective. By contrast, we developed an environment in which goal selection was the main dependent variable, enabling us to study key determinants of goal setting in humans. The experiment was implemented in the Unity game engine and presented as an interactive online game, in which healthy human participants (N = 175; see Appendix A for details) played the role of alchemists and could learn to make different potions from sets of ingredients. Below we refer to potions as "goals" and ingredients as "actions".

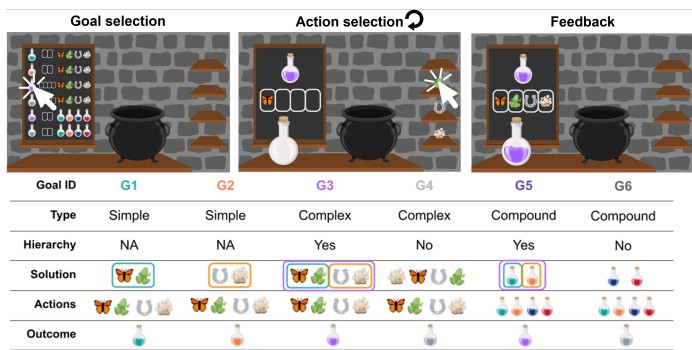

Figure 1: Structure of learning trials and summary of the goal space. **(Top)** An example trial. **(Bottom)** Schematic representation of goals and their characteristics.

The paradigm included training, learning, and testing phases. Learning phase trials (144) were composed of a goal selection stage, an action selection stage, and a feedback stage (Figure 1, top; Appendix B). In the goal selection stage, participants viewed colored potions available for selection, with icons indicating the number of ingredients required and the four different ingredients available to make them. In the action selection stage, participants chose which ingredients to add to the cauldron. In the feedback stage, participants received binary feedback as a result of their actions (the empty flask in the foreground either filled up or remained empty based on whether participants chose the correct ingredients in order). In the 12 training trials preceding the learning phase, goal selection

was forced to ensure participants experienced each goal at least twice. In the final testing trials, participants were prompted to try and make each potion 4 times without feedback.

Potions could either be made of basic ingredients ("simple" and "complex" goals G1-G4) or potions of different colors ("compound" goals G5-G6) and required either two (simple and compound goals G1-G2, G5-G6) or four actions to complete (complex goals G3-G4; Figure 1, bottom). Two potions could be made from either basic ingredients or other potions (potions for G3-G6). One complex goal ("hierarchical", G3) was made by combining the ingredients of the two simple potions (G1, G2), whereas the other complex goal was not ("non-hierarchical", G4). One complex goal ("hierarchical", G5) was made by flasks of the two simple potions (G1, G2), whereas the other complex goal was not ("non-hierarchical", G6). Thus, G3 and G4, and G5 and G6, were matched in terms of actions required to create the corresponding potions but differed in whether knowledge from other potions could be used to find the corresponding solutions faster. Two additional, simple goals made of the first two and last two ingredients of G4 were presented to participants in the testing phase (G7 and G8; Appendix B). Unless otherwise specified, we focus on goals G1-G6, which participants engaged with in the training and learning phases. Further details about the goal space are provided in Appendix B. The information displayed in Figure 1 was never shown to participants. The correspondence between goal identity, color, position on the screen, and solution was pseudo-randomized across participants.

Various features make the experimental design well-suited to study the contributions of LLP to goal selection by helping decorrelate LP from LLP. First, the correct response was static and signaled through *deterministic feedback*. While the standard LP measure only flattens upon repeated subsequent attempts, deterministic feedback enables LLP to reach its limit after a single correct response. Second, the *action space is limited* and *solutions are unique*, so negative feedback can be used to exclude the attempted action sequence from the set of possible solutions. This feature was especially meaningful for studying LLP, which can be immediately updated following failed attempts with a novel action sequence. Third, the choice space is vast enough that *rewards are sparse* early on in learning, which further decorrelates LP and LLP since LP is 0 until the first successful attempt, while LLP is updated at the first trial no matter the outcome. Fourth, *hierarchical relationships* exist among some, but not all goals. Upon becoming aware of hierarchical relationships, participants could take shortcuts to identify the correct recipe for the goals involved, leading to sharp changes in LLP even with small differences in performance. This feature additionally enables us to test for hierarchy effects on goal selection beyond the indirect effects of hierarchy on learning through LLP.

### 3.2 Computational modeling

On each trial, participants first selected a goal and then a sequence of actions. Unless otherwise specified, we only modeled goal selection, conditioning on true participants' actions within each trial after goal selection. However, action selection influences performance, LP, and LLP. As in existing studies [23], we model human goal selection as a multi-arm bandit problem, where the probability of selecting a goal depends on its subjective value relative to other goals. Multiple factors $f$ may jointly contribute to the overall subjective value of a goal [23, 24], which is thus calculated as a weighted sum of the goal's value relative to each factor $V_f$ times its weight $\beta_f$. The probability $P^t(g^*)$ of choosing goal $g^*$ among possible goals $G$ on trial $t$ is obtained through a softmax function over the goal values (as is standard in the decision-making literature [86]):

$$P^t(g^*) = \frac{exp(\sum_f \beta_f \cdot V_f^t(g^*))}{\sum_{g \in G} exp(\sum_f \beta_f \cdot V_f^t(g))}$$

Subjective goal values are updated as a function of experience through the delta rule [87]:

$$V_f^{t+1}(g^*) = V_f^t(g^*) + \alpha \cdot \delta_f^t(g^*)$$

where $\alpha$ is a learning rate for value updates and $\delta_f$ is factor-dependent. Models differed in which factor, or combination of factors, composed the overall goal value. Below we detail the individual factors we considered.

**Performance**   On each trial, the utility of the selected goal with respect to performance is updated based on the goal-contingent feedback received on that trial $r^t$ (1 for positive feedback, 0 for negative feedback):

$$\delta_{\text{performance}}^t(g^*) = r^t - V_{\text{performance}}^t(g^*)$$

Performance estimates were initialized as $V_{\text{performance}}^0(g^*) = \frac{1}{6}$ where 6 is the number of goals.

**Learning progress**  Learning progress (LP) was updated through the change in reward prediction error for the current goal after initializing $V_{\text{LP}}^0(g^*) = 0$ (see [24] for a similar approach) . "Reward", here refers to the feedback $r$ participants received on each trial. However, note that no external rewards were delivered for successfully obtaining the desired potion (Appendix C). On each trial, the reward prediction error was calculated as the difference between the current performance value estimate and the feedback obtained. The reward prediction error on trial $t$ was compared to the reward prediction error on the previous trial $t-1$:

$$\delta_{\text{LP}}^t(g^*) = [(r^t - V_{\text{performance}}^t(g^*)) - (r^{t-1} - V_{\text{performance}}^{t-1}(g^*))] - V_{\text{LP}}^t(g^*)$$

**Latent learning progress**  Goal utilities with respect to latent learning progress (LLP) were updated based on the difference between the estimate at trial $t$ and the current trial's LLP:

$$\delta_{\text{LLP}}^t(g^*) = LLP_{g^*}^t - V_{\text{LLP}}^t(g^*)$$

with $V_{\text{LLP}}^0(g^*) = 1$. Until the goal was learned (when LLP immediately became 0) subjective (unobservable) estimates of LLP were approximated as $LLP_{g^*}^t = 1 - \frac{N_{\text{action sequences tested}}}{N_{\text{possible action sequences}}}$. The number of possible action sequences is 12 for 2-action goals and 24 for 4-action goals. The number of action sequences tested is the sum of unique action sequences the agent has tried up until trial $t$. A goal is considered learned when the subject has obtained positive feedback for the current goal at least once (but see Appendix D for alternative heuristics and approximations). Therefore, this formulation of LLP captures the size of the space left to explore to find the correct sequence of actions for the current goal, i.e. the "distance" from the solution. As a result, negative $\beta_{LLP}$ values indicate a high willingness to pursue activities one is close to solving. This choice of formulation – rather than its opposite, i.e. one where high values indicate high levels of progress toward perfect performance – was made to maintain continuity in the LLP function, which is 0 after a solution is found.

**Hierarchy**  Hierarchical relationships between goals could impact goal selection in at least two ways. First, indirectly, by enabling inferences about the solution for goals involved in the hierarchy, thus speeding up exploration and changes in LLP. Second, directly, by acting on biases people might have for selecting goals that share common structures. To account for the latter, we also included a hierarchy factor, whose estimates were updated via:

$$\delta_{\text{hierarchy}}^t(g^*) = H_{g^*} - V_{\text{hierarchy}}^t(g^*)$$

with $V_{\text{hierarchy}}^0(g^*) = 0$ and $H$ equal to 1 for hierarchical (G3, G5), -1 for non-hierarchical (G4, G6), and 0 for simple goals (G1, G2; but see Appendix D for an alternative scheme and update rule).

**Model space**  Because performance is a powerful motivator in people's decisions about time allocation [23, 45], we include it in all our models. In addition to a performance-only model, we fit and compared models with additional factors for LP, LLP, hierarchy, and both LLP and hierarchy. Thus, fit parameters included a shared $\alpha$ across factors and weighting parameters $\beta_f$ for each factor in the model. Adding a control variable for choice perseveration improved the likelihood of goal choices but did not affect modeling results (Appendix D), so we removed it to facilitate interpretation.

**Model fitting and comparison**  Models were fit and compared through hierarchical Bayesian inference (HBI; [88]) based on the likelihood of participants' goal selection choices [18, 58]. We compare models through protected exceedance probability (PXP), i.e., the probability that each model is the most frequently expressed in the studied population after accounting for chance [89], and average responsibility (R), which captures how well each model explains participants behavior. Models and parameters for the winning model were recoverable [90] (Appendix E). Simulating participants' behavior [91] also required modeling the learning process, which we assumed followed a reinforcement learning architecture with options [92] (Appendix F).

## 4    Behavioral results

Given the novelty of our experimental paradigm, it was crucial to validate the environment and gather intuitions about participants' behavior. Furthermore, both LP and LLP are influenced by how well participants learn to achieve each goal. We thus first analyze group-level learning and goal selection as a function of goals' difficulty levels and hierarchical structures. We then test the specific role of LLP in goal selection through computational modeling (Section 5).

**Participants learn successful goal-conditioned policies** Learning phase performance was above chance for each goal (Wilcoxon tests were used in place of parametric t-tests since the data was not normally distributed according to Shapiro-Wilk tests of normality; chance levels were 0.083 for 2-action and 0.042 for 4-action goals; all average performances $\leq$ 0.29, all W(174) $\geq$ 160, all p $\leq$ 0.004; Figure 2A). As expected, performance was higher for 2-action (M = 0.51 $\pm$ 0.02) than 4-action goals (M = 0.33 $\pm$ 0.02, W(174) = 1207, p < 0.001, r = 0.51). Similar patterns for goals G1-G6 were observed in the testing phase, where performance was not contingent on goal selection (Figure 2B).

**Hierarchical inference supports learning** If individuals recognized hierarchical relationships during learning, they could use them to draw inferences about the correct recipes for the potions involved. For instance, upon discovering that G5 could be made from the potions created in G1 and G2, participants may correctly infer that G3 is achieved by chaining the recipes for G1 and G2 (Figure 1, bottom). Consistent with the hypothesis that at least some participants exploited the environment's hierarchical structure, learning performance was higher for hierarchical goals compared to their non-hierarchical counterparts (complex: hierarchical (G3) M = 0.37 $\pm$ 0.02, non-hierarchical (G4) M = 0.29 $\pm$ 0.02, W(174) = 4168.5, p = 0.004, r = 0.16; compound: hierarchical (G5) M = 0.53 $\pm$ 0.02, non-hierarchical (G6) M = 0.49 $\pm$ 0.02, W(174) = 6036.5, p = 0.032, r = 0.11; Figure 2A). Similarly, participants took fewer attempts to learn solutions to hierarchical than non-hierarchical goals (complex: hierarchical (G3) M = 16.66 $\pm$ 1.19, non-hierarchical (G4) M = 22.1 $\pm$ 1.23, W(120) = 1884.5, p = 0.009, r = 0.17; compound: hierarchical (G5): M = 11.66 $\pm$ 0.58, non-hierarchical (G6) M = 13.21 $\pm$ 0.53, W(155) = 3889.5, p = 0.03, r = 0.12; Figure 2C). However, hierarchy effects were not present in the testing phase, likely due to ceiling effects (complex: hierarchical (G3) M = 0.56 $\pm$ 0.03, non-hierarchical (G4) M = 0.52 $\pm$ 0.03, W(174) = 1592.5, p = 0.223, r = 0.07; compound: hierarchical (G5) M = 0.73 $\pm$ 0.03, non-hierarchical (G6) M = 0.73 $\pm$ 0.03, W(174) = 1158.5, p = 0.765, r = 0.02; Figure 2B).

**Goal selection is sensitive to performance and hierarchy** In assessing whether difficulty levels and hierarchy affected participants' goal selection, we found it displayed opposite patterns to learning performance (Figure 2D). The average probability of selecting a goal was lower for 2-action (M = 0.15 $\pm$ 0.00) than 4-action goals (M = 0.19 $\pm$ 0.01; W(174) = 4214, p < 0.001, r = 0.27). Overall, participants chose the hierarchical (G3; M = 0.18 $\pm$ 0.01) less than the non-hierarchical complex goal (G4; M = 0.21 $\pm$ 0.01, W(174) = 5231.5, p = 0.002, r = 0.17). Differences between hierarchical (G5; M = 0.15 $\pm$ 0.01) and non-hierarchical compound goal selection were not significant (G6; M = 0.16 $\pm$ 0.00, W(174) = 6444, p = 0.128, r = 0.08). At the individual level, the greater the

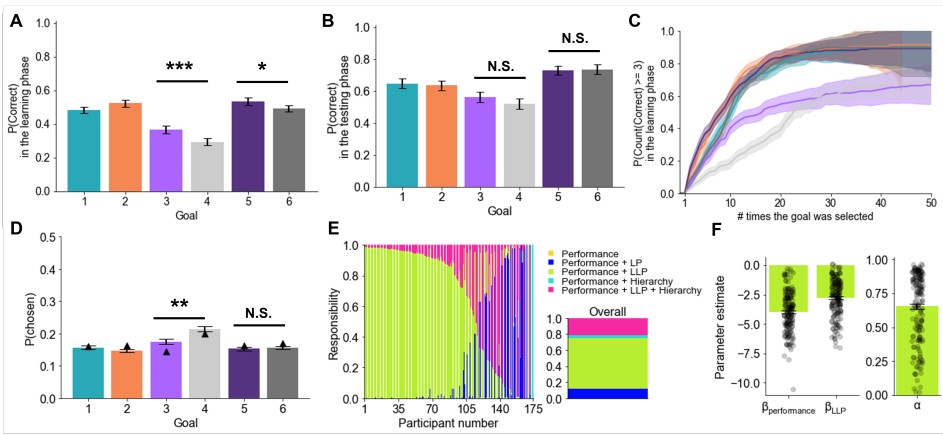

Figure 2: Behavioral and modeling results. **(A)** Learning performance was above chance and showed hierarchy effects. **(B)** Test performance was better than chance, but no significant hierarchy effects were detected. **(C)** On average, fewer attempts were needed to learn hierarchical (G3) compared to non-hierarchical (G4) 4-action goals. **(D)** Partial hierarchy effects are present in goal selection. The winning model (triangle markers) reproduces this pattern. **(E)** Model responsibilities for individual participants and the overall studied population. **(F)** Best-fitting parameters (HBI) for the winning model. Bars and shading indicate the SEM, dots individual participants. *** p < 0.001, ** p < 0.01

difference in performance between 2-action and 4-action goals, the lower the probability of selecting a 2-action goal (Spearman's $\rho$ = -0.35, p $<$ 0.001), and the greater the performance difference between hierarchical and non-hierarchical goals, the lower the probability of selecting a hierarchical goal (complex: $\rho$ = -0.68, p $<$ 0.001; compound: $\rho$ = -0.51, $<$ 0.001). These results indicate that goal selection depends on performance and hierarchical relationships between goals. We next turn to computational modeling to detail the contributions of performance, hierarchy, LP, and LLP to goal selection.

## 5 Modeling results

We that predicted LLP supports human goal selection, thus expecting models relying on this signal to fit participants' goal selection data better than models that do not.

**Latent learning progress guides goal selection**   The model with performance and LLP utility factors was the most strongly represented across participants (PXP = 1; R = 0.63) and replicated the main goal selection patterns seen in participants' data (Figure 2E). Some participants were also well-captured by a model that additionally integrated hierarchy into goal values (R = 0.22). We note that hierarchy likely plays a more direct role in learning (thereby impacting LLP) than in goal selection, given that hierarchy effects in goal selection and performance could be recovered after eliminating hierarchical components from goal selection, but not learning (Appendix G). The model selecting goals based on performance and LP did not account well for participants' behavior (R = 0.12). Models with just performance (R = 0.00) or performance and hierarchy factors (R = 0.04) were marginally represented. The winning model's best-fitting $\beta_{\text{performance}}$ and $\beta_{\text{LLP}}$ were on average negative, suggesting most individuals focused on activities where performance was low – indicating the solution had yet to be found – and where LLP was low (Figure 2F) – indicating learning potential was high (Section 5). However, more nuanced patterns and temporal dynamics may exist at the individual level.

## 6 Inter-individual differences in learning and goal selection

Several interesting strategies emerged upon inspecting individual behaviors (Figure 3). We first exemplify them before using a data-driven approach to classify participants according to their approach to the game, revealing differences in how hierarchy impacts learning and thus goal selection. Finally, we assess whether LLP supports goal selection across groups.

**People express a variety of goal and action selection strategies**   Many participants repeatedly selected the same goal – often testing possible solutions in a systematic order (Figure 3A) – only switching to a new goal after learning the current one (Figure 3A,B). After learning to make all potions, some participants began alternating goals, as if testing or training their ability to retain multiple solutions (Figure 3A,B). Although most participants showed interest in learning all goals (Figure 3A,B), a small subset of individuals preferred setting few objectives, or even a single goal, in which they excelled (Figure 3C), while disregarding others. Lastly, some participants moved across goals and ingredients in a seemingly unprincipled manner, presumably because they were confused or unmotivated (Figure 3D). A subset of the participants seemed to be aware of adopted strategies (see Appendix H for example verbal reports). We next clustered participants to explore how idiosyncrasies in learning related to goal selection. Note that participants in Figure 3 were chosen as examples and do not directly relate to the clustering analyses presented below.

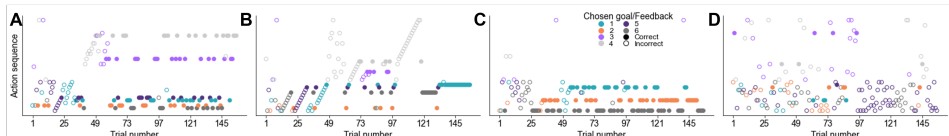

Figure 3: Example behaviors from four participants. **(A-D)** Chosen action sequence as a function of trial number (training and learning phases), color-coded by goal and feedback.

**Clustering of individual strategies reveals differences in hierarchical learning** We used K-means clustering [93] to identify 4 groups of participants, where K was determined through the elbow method (Appendix I). The behavioral metrics used to identify the clusters (e.g., performance and goal selection entropy and distributions) can be found in Appendix I. To avoid statistical double-dipping [94], we limit ourselves to qualitative observations on cluster-based behavior. To better illustrate salient characteristics of each cluster, we also identified "archetypal" participants through archetypal analysis – a method, often used to profile video game players [95], which detects data points that can reproduce the original population when linearly recombined [96]. We then matched each archetypal feature with the best-aligned cluster (Appendix I) and identified the participant closest to the corresponding archetype as an extreme example of the cluster it belonged to (Figure 4M-P).

Participants in cluster 1 (N = 39) had below-average performance at learning (M = 0.18 ± 0.03) and testing (M = 0.27 ± 0.06) and favored 2-action goal. They often failed to learn or retain solutions and switched goals frequently (Figure 4M). In the largest cluster 2 (N = 75; Figure 4B), performance was excellent at learning (M 0.59 ± 0.0) and testing (M = 0.86 ± 0.03). This group leveraged hierarchical structures to speed up learning (Figure 4B,F) – which also impacted goal selection. Non-hierarchical goals were favored (Figure 4J) and often learned later than hierarchical ones (Figure 4N). Cluster 3 participants (N = 53) showed relatively high performance at learning (M = 0.26 ± 0.04) and testing (M = 0.62 ± 0.06; (Figure 4C), but seemed to struggle with all 4-action goals – performing worse in them but choosing them more – and took less advantage of hierarchical relationships across goals (Figure 4G,K,O). Given its small size (N = 8), we invite caution in interpreting the results for cluster 4. However, this group showed an interesting, strong tendency to repeatedly select the same, small set of goals at which they succeeded (often preferring 2-action goals; Figure 4L,P). As a result, their performance was excellent in the learning phase, where goals were freely chosen (M = 0.67 ± 0.13, Figure 4D,L) but dropped in the test phase, where goals were imposed (M = 0.46 ± 0.16).

Despite such distinct patterns of behaviors, LLP was a central component of goal selection in all clusters (Figure 4Q-T), suggesting LLP is a flexible signal that adapts to idiosyncrasies in people's learning while similarly guiding goal selection (but see Appendix G for further discussion of how learning interacts with goal selection). At the same time, the best-fit parameters of each cluster captured specific properties of the different groups (Figure 4Q-T insets). The efficient learners of cluster 2 relied more strongly on LLP (cluster 2 $M_{\beta_{LLP}}$ = -3.17 ± 0.18, other clusters $M_{\beta_{LLP}}$ -2.46 ± 0.16, Mann-Whitney U(173) = 2696, p = 0.001, r = 0.24) and updated both performance- and LLP-based goal values faster than other groups (cluster 2 $\alpha$ = 0.73 ± 0.03, cluster 2 $\alpha$ = 0.6 ± 0.03, U(173) = 4979, p < 0.001, r = 0.28). Cluster 3, which seemed less likely to rely on hierarchy, was more attuned to performance when selecting goals (cluster 3 $M_{\beta_{performance}}$ = -4.63 ± 0.22, other clusters $M_{\beta_{performance}}$ = -3.67 ± 0.16, U(173) = 2274, p = 0.002, r = 0.24)). Cluster 1, which lacked precise strategies, was less sensitive to both performance (cluster 1 $M_{\beta_{performance}}$ = -2.93 ± 0.26, other clusters $M_{\beta_{performance}}$ -4.25 ± 0.14, U(173) = 3778, p < 0.001, r = 0.31) and LLP (cluster 1 $M_{\beta_{performance}}$ = -1.74 ± 0.16, other clusters $M_{\beta_{performance}}$ -3.06 ± 0.14, U(173) = 3980, p < 0.001, r = 0.36).

# 7 Discussion

Human behavior is often motivated by self-defined challenges. Despite the centrality of goals in human learning [19], how people select their own objectives without external guidance or incentives remains to be specified. Following advances in artificial open-ended skill learning [26–28], learning progress (LP) has emerged as a useful signal for human goal selection [23, 24]. Here, we introduced a "latent" form of LP (LLP), which is informed by an agent's model of the environment and memory of the actions pursued in it, in addition to the manifest changes in performance tracked by standard LP. In a purposefully developed hierarchical reinforcement learning task, we find that people's goal selection is driven by LLP, calling for an extension of the LP notion to include its latent variants.

Inter-goal relationships in our test environment enabled us to check whether hierarchy additionally drives goal selection, as intuition suggests (e.g., one might decide to become competent in baking by first relying on ready-made dough and only later setting the lower-level goal of making dough from scratch). We find that hierarchy impacts goal selection indirectly by enabling inferences and thus affecting LLP. That is, participants' learning is sensitive to hierarchical structures, leading to changes in LLP which, in turn, affect goal selection. We also gathered partial evidence for a standalone role of hierarchy in curriculum development, which further studies may address.

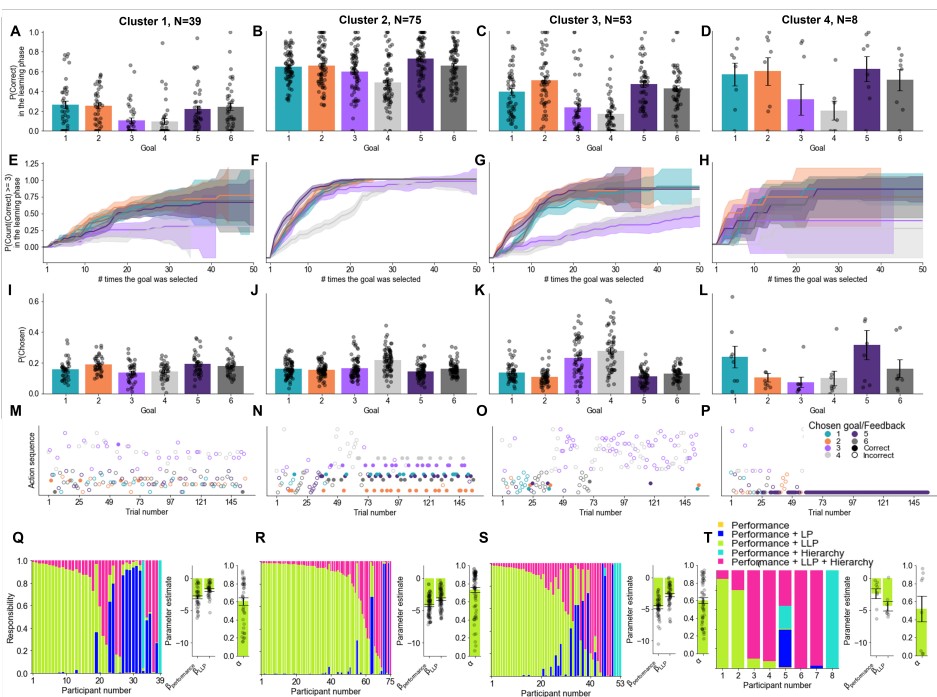

Figure 4: Behavioral and modeling results by cluster (1-4, left to right). **(A-D)** Learning performance. **(E-H)** Probability that each goal was learned over the number of times it was selected. **(I-L)** Goal selection probabilities. **(M-P)** Behavior of each cluster's closest-matching archetypal participant. **(Q-T)** Model responsibilities and best-fitting parameters.

We found marked inter-individual differences in strategies for goal setting and pursuit. However, our sample was restricted to a relatively homogeneous group of undergraduate university students. Future studies and computational models may extend participation to a broader subject pool and specifically address individual differences in goal selection and achievement, as well as their relationship with demographics and cultural background, cognitive abilities, and psychopathology (cf. [97]), which might inform personalized teaching strategies and productivity tools. As our knowledge of human goal setting becomes more precise, however, ethical concerns regarding the use of behavioral sciences in marketing and management should be addressed, particularly in cases where highly personalized methods of influence could transform advertising into manipulation.

Elucidating the computational mechanisms driving human goal selection may propel advances in open-ended machine learning. Our suggestion, wherever relevant, is to incorporate LLP in artificial autotelic agents' goal selection. Compared to classic LP, LLP is sensitive to latent changes in an agent's abilities – which LP would only be able to capture after repeatedly observing positive feedback. Using LLP could result in a faster adaptation of the agent's goal selection, ultimately speeding up its learning by enabling it to detect progress before receiving the first reward and letting it proceed to more challenging tasks sooner.

Some limitations of the present study may need to be addressed before other fields can fully benefit from the results we presented. In our task, people chose goals from a predefined menu of options. Although this facilitates the study of goal setting, people often invent their own goals by combining observations and imagination [98]. For simplicity, our modeling focused on goal selection. Future research, however, should jointly model goal choices and goal-conditioned action selection, since – as our data suggests – learning participates in complex interactions with the goal selection process. While we provide a simple and task-specific formalization of LLP, a generalized definition is necessary to understand the differences between LLP and other intrinsic motivation signals and ease the implementation of LLP-based goal selection in autotelic machines. By providing initial evidence for LLP, we hope to inspire the establishment of even more precise signals for goal selection in both humans and artificial agents.

## Acknowledgments

In producing this work, G.M. was supported by the Chateaubriand Fellowship of the Office for Science & Technology of the Embassy of France in the United States; P-Y.O. was supported by ANR AI Chair DeepCuriosity. A.G.E.C. was supported by NIH Grant 1R01MH119383-01.

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

## Appendix overview

To help the reader, we provide an overview of the contents covered by the ensuing appendices:

A. Participants: Information about the human participants who took part in the experiment.

B. Experimental paradigm details: Detailed information about the protocol and the learning paradigm's environment.

C. Experiment instructions: Instructions provided to participants at the start of the experiment.

D. Modeling robustness checks: Supplementary analyses to verify the robustness of our modeling results in the face of small modifications to the model designs.

E. Parameter and model recoverability: Results of the parameter and model recoverability studies.

F. Options reinforcement learning model for simulations: Description of the design and use of models with a learning component (used for simulations only).

G. The role of hierarchy in learning and goal selection: Results and discussion of some evidence for a role of hierarchy learning in the context of the present experimental paradigm, and how it relates to goal selection.

H. Verbal reports of example participants' strategies: Example verbal reports from participants about their adopted learning strategies.

I. Clustering methods: Details about the clustering methods.

## A  Participants

One hundred and seventy-nine participants were recruited through the University of California, Berkeley, Research Participation Program pool of participants and received course credit as compensation for their time. Participants completed the experiment online from their own devices. We excluded four participants who either responded affirmatively when directly asked whether they took written notes about the correct recipe for each potion, or mentioned writing down solutions on their phone or a piece of paper when reporting their strategies in a follow-up questionnaire. Thus, 175 participants (76% female, age: range = 18-47, M = 21.21 $\pm$ 3.75, 94% right-handed) were included in the final data set. All were native or fluent English speakers with no medical history of brain injury, mental/psychiatric illness, or drug/alcohol abuse. The Institutional Review Board at the University of California, Berkeley, approved the study.

## B  Experimental paradigm details

### B.1  Gameplay

The paradigm was divided into training, learning, and testing phases. Each learning phase trial encompassed a goal selection, an action selection, and a feedback stage. In the *goal selection stage*, participants viewed a blackboard depicting six "goal" potions, characterized by their color. Next to each potion were either two or four empty slots – indicating the number of ingredients required to make the potion – and illustrations of the four different ingredients available for that potion. Potions could either be made of basic ingredients (a frog, a bunch of mushrooms, a horseshoe, and a butterfly), or potions of different colors (e.g., teal, orange, red, and blue-colored potions). Two potions appeared twice on the left side of the board, as they could be made from either basic ingredients or other potions. Participants were prompted to select one of the available potions on the left side of the blackboard with a mouse click (Figure 1). At the beginning of the *action selection stage*, the blackboard contents changed to depict the goal potion and either two or four empty slots, which indicated the number of ingredients required. Four different ingredients appeared on separate shelves on the right side of the screen. When a participant clicked on an ingredient, it moved from the shelves to a cauldron depicted in the center of the screen and appeared in the corresponding slot on the blackboard. An empty flask appeared on the table next to the cauldron for the duration of the action selection stage. Participants were prompted to select ingredients, one at a time, until all slots were filled (Figure 1, top). During the *feedback stage*, the flask either remained empty and shook left and right, signaling failure to make

the goal potion, or became filled with the same color as the goal potion and wiggled up and down, signaling that the chosen ingredients were added to the cauldron in the correct order. No points or otherwise explicit feedback messages were provided.

The learning phase (144 trials) was divided into 6 blocks, at the beginning of which participants were allowed to take a break if needed. The training phase (12 trials, 2 per goal) served to help participants get familiar with the game flow and was similar to the learning phase, except that goal selection was forced. This section comprised 12 trials (two per goal), and was intended to help participants get familiar with the game flow. The testing phase was similar to the learning phase, except that goal selection and feedback stages were skipped. Here, participants were instead prompted to make a given potion without receiving information about whether the selected order and ingredients were correct. A set of instructions preceded the gameplay (Appendix C). Importantly, participants were not told to try and learn all the correct mappings between actions and goal achievement. We did not disburse performance bonuses, nor were there any external incentives for successful goal completion. Instead, participants were free to explore the goal space as they pleased for a fixed number of trials.

At the end of the experiments, participants completed a short demographics questionnaire and optionally answered the questions "Did you try to use any strategy while playing the game? If so, what was it?", and "Did you notice any structure in the game? If so, what did you notice?". A subset of participants were also asked, "Did you have specific aims or objectives while playing? If so, what were they?". Participants were also invited to leave written feedback.

## B.2    Goal space

The six different goals (potions) available for selection were constructed as illustrated in Figure 1, bottom. While we present them in an orderly fashion to ease readability, participants were not told about the underlying structure of the set of goals. The relative position on the screen of goals with different features was pseudorandomized across participants. Below is a description of the six available goals from the training and learning phases:

- G1: simple goal 1. This potion was made of two basic ingredients.
- G2: simple goal 2. This potion was made of two basic ingredients, none of which were used to make G1.
- G3: hierarchical complex goal. This potion was made of four basic ingredients, in an order that reflected the recipe for G1 followed by that for G2. Thus, participants who became aware of the structure could leverage knowledge of G1 and G2 to learn G3, or vice-versa.
- G4: non-hierarchical complex goal. Like G3, this potion was made of four basic ingredients. However, the order in which they had to be added to the cauldron did not reflect G1 or G2 in any way. Thus, learning G1 or G2 did not offer any learning advantage for G4, or vice-versa.
- G5: hierarchical compound goal. This potion had the same color as G3. However, potions of different colors were available for making G5, as opposed to the basic ingredients available for G4. The potions required to make G5 correctly were G1 followed by G2. Learning this potion could, in theory, be used to gain insight into how G3 is made, or vice-versa.
- G6: non-hierarchical compound goal. This potion had the same color as G4 but, similar to G5, was made of two other potions rather than basic ingredients. These were potions that were not available as goals, such that learning G6 could not leverage or provide insight into other potions' recipes. However, participants could infer that the recipes for the first and second potions used to make G6 were the first two and last two ingredients used in G4.

Two additional goals were presented to participants in the testing phase:

- G7: simple goal 3. This potion was made of the first two basic ingredients of G4.
- G8: simple goal 4. This potion was made of the last two basic ingredients of G4.

## C    Experiment instructions

We report the instructions provided to participants verbatim below. Each item corresponds to a screen presented to participants, who paced their own reading by moving to the next screen upon pressing

the spacebar key on their keyboard. They were instructed to do so by the message "Press SPACEBAR to continue" appearing at the bottom of each screen. Wherever applicable, images accompanying each item are references in the following list.

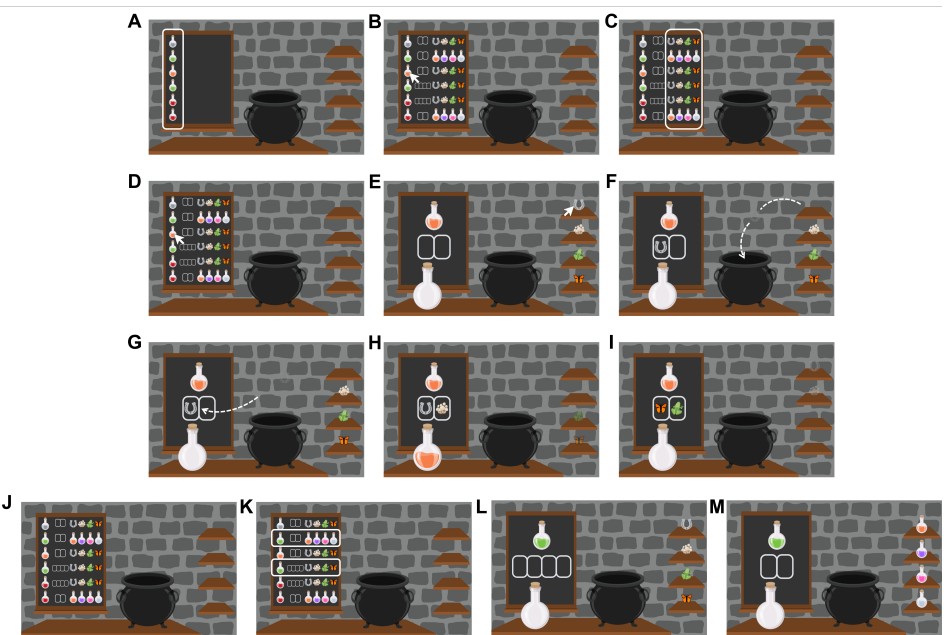

Figure 5: Images accompanying the experiment instructions.

1. Welcome to our study! Thank you very much for deciding to participate.

2. This experiment requires your full attention. We will only be able to use your data if you participate fully in the experiment, which should only take about 35 minutes when you are fully engaged.

3. Before we begin, please maximize your browser window and turn off all notifications (Facebook, Instagram, texts, etc.) on both your phone and computer.

4. Please do not disengage (for example, by switching tabs in your browser or navigating away from the task) while you take part in the experiment, including during breaks.

5. If you're not engaged, your data are not valid and cannot be used for the experiment, and therefore you may not get credit. Thank you for understanding.

6. In this game, you will play the role of an alchemist and try to make potions from a set of ingredients.

7. On each trial, you will see a blackboard that shows all the potions you can try to make. [Figure 5A]

8. Note that this is just an example setup, and things may look slightly different in the actual game.

9. Each potion is made of either 2 or 4 ingredients. The empty slots next to each potion indicate how many ingredients that potion requires. [Figure 5B]

10. The images shown on the far right of the blackboard illustrate which ingredients you can use to try and make each potion. [Figure 5C]

11. In the game, you will first select a potion by clicking on it. [Figure 5D]

12. Once you have chosen a potion to make, you will select one ingredient at a time from the shelves on the right to try and make it. [Figure 5E]

13. Once you click on an ingredient, it gets added to the cauldron. [Figure 5F]

14. The blackboard will help you keep track of which ingredients you have selected and which potion you are trying to make. [Figure 5G]

15. Each ingredient can only be selected once, and the order in which they are added matters.

16. If you select the correct ingredients in the correct order, you will see the empty flask on the table fill up with the potion you selected. [Figure 5H]

17. If the ingredients were incorrect or not chosen in the right order, the flask will remain empty. [Figure 5I]

18. After you have clicked on the required number of ingredients, you will get to choose a potion again. You are welcome to try and make the same potion again or choose a different one on each trial. [Figure 5J]

19. Some potions can be made in two ways. [Figure 5K]

20. Either by combining basic ingredients: [Figure 5L]

21. Or by combining other potions: [Figure 5M]

22. Things may seem confusing at first, but the game will become clearer as you keep trying!

23. To help you familiarize yourself with the game, for the first few trials the computer will choose potions for you to make.

24. Are you ready? Press SPACEBAR to begin.

## D    Modeling robustness checks

We performed several checks to verify our modeling results were robust to small variations in the adopted methods. To save time and resources in these additional analyses, we compare models through exceedance probability (XP) rather than the more computationally expensive PXP [89].

### D.1    Adding a perseveration term to all models

Since adding a perseveration term can benefit models of human choices [99], we also ran a version of the model comparison pipeline in which each model included such a feature in the goal valuation process. The utility $V_{\text{perseveration}}$ grew each time a goal was selected to the extent determined by $\alpha$. Adding perseveration to each model did not change the results (winning model XP = 1; R: model with perseveration only = 0.00, perseveration and performance = 0.00, perseveration, performance, and LP = 0.12, perseveration, performance, and LLP = 0.70, perseveration, performance, and hierarchy = 0.06, perseveration, performance, LLP, and hierarchy = 0.17). For simplicity and ease of interpretation, we thus opted for models without perseveration.

### D.2    Using 1 vs. 3 successful attempts to define when a goal is learned

Our implementation of LLP drops to 0 after a participant has learned the solution to a goal, as (assuming perfect memory) no further learning progress can be made once the solution is found. Since we did not have access to the true cutoff for when a goal was "learned", we placed an arbitrary cutoff after the first time the solution was identified. However, participants may require more than a single trial to retain a goal's correct action sequence. Therefore, we also developed models where LLP dropped to 0 after 3 successful attempts. Using this heuristic instead of the 1 successful attempt heuristic did not change the results (winning model XP = 1; R: model with performance = 0.03, performance and LP = 0.18, performance and LLP = 0.66, performance and hierarchy = 0.00, performance, LLP, and hierarchy = 0.13).

### D.3    Using a Bayesian cutoff to define when a goal is learned

To test the reliability of a heuristic method to decide after how many trials a goal was "learned" for the purpose of updating LLP, we also used a Bayesian cutoff to set a more precise threshold. This was calculated by computing confidence intervals over each subject's learning curve and determining the trial at which performance surpassed chance levels with 95% confidence (see [100] for details on methods). Using the Bayesian threshold for learning did not change the results (winning model XP = 1; R: model with performance = 0.00, performance and LP = 0.14, performance and LLP = 0.62, performance and hierarchy = 0.04, performance, LLP, and hierarchy = 0.21).

### D.4 LLP as the proportion of possible solutions tested

Within the specifics of our task, LLP decreased every time a participant chose a new action sequence for the active goal, eventually reaching 0 once the participant found the solution. This ensured there were no discontinuities in LLP. However, another intuitive way to model LLP is by incrementing it every time a participant attempted a new action sequence, until they found the solution:

$$LLP_G^t = \begin{cases} \frac{N_{\text{action sequences tested}}}{N_{\text{possible action sequences}}} & \text{if } G \text{ is not learned} \\ 0, & \text{otherwise} \end{cases}$$

Since this formulation did not significantly change the results (winning model XP = 1; R: model with performance = 0.00, performance and LP = 0.19, performance and LLP = 0.59, performance and hierarchy = 0.36, performance, LLP, and hierarchy = 0.18), we kept the version described earlier, which was not affected by abrupt discontinuities – thus facilitating interpretation.

### D.5 Alternative coding scheme for the hierarchy feature

To update the hierarchy utility of the chosen goal, we used a term $H$ equal to 1 for hierarchical (G3, G5), -1 for non-hierarchical (G4, G6), and 0 for simple goals (G1, G2). However, hierarchy effects were more prominent for complex than compound goals (Figure 2). We thus ran the model comparison pipeline with an alternative coding scheme for $H$, which was equal to 1 for the hierarchical complex goal (G3), -1 for the non-hierarchical complex goal (G4), and 0 otherwise (G1, G2, G5, G6). Using this scheme did not significantly change the results (winning model XP = 0.96; R: model with performance = 0.02, performance and LP = 0.12, performance and LLP = 0.48, performance and hierarchy = 0.02, performance, LLP, and hierarchy = 0.36).

### D.6 Alternative coding scheme for the hierarchy feature

In our model, the term $H$ was used to learn the utility of each goal with respect to hierarchy over time through the shared learning parameter $\alpha$. However, it is also possible that hierarchy information informs participants' behavior from their first interaction with each goal. We thus ran the model comparison pipeline with a learning rate of 1 for $H$, equivalent to setting hierarchy values to their coding scheme from the first interaction with a goal. Using this fixed learning rate did not significantly change the results (winning model XP = 1; R: model with performance = 0.03, performance and LP = 0.12, performance and LLP = 0.58, performance and hierarchy = 0.00, performance, LLP, and hierarchy = 0.27).

## E Parameter and model recoverability

Before analyzing model parameters from the winning model (one which chose goals based on performance and LLP signals), we simulated and fit data with known parameters (best-fit parameters from the Laplacian approximation of HBI) to ensure they could be recovered. We used an options reinforcement learning model to simulate participants' learning process (Appendix F). We observed tight correlations between true and recovered parameters (Figure 6), suggesting the experiment was sufficiently powered to assess this model [90].

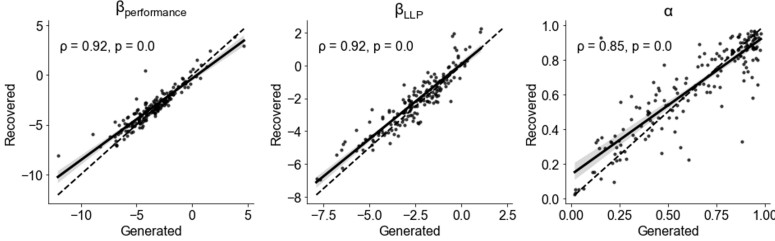

Figure 6: Parameter recovery. Spearman's correlations between true and recovered parameters for the winning model.

To ensure we could compare similar models [90], we tested model recoverability by simulating participants' behavior with each model and the corresponding best-fit parameters (Laplacian approximation step of HBI). Each participant's behavior was simulated 3 times to account for random noise. We again assumed a a reinforcement learning process with options governed the participants' action selection. We then performed model comparison (HBI; [88]) to determine whether we could correctly identify the simulating parameters. The confusion matrices in Figure 7 illustrate this was the case, although the model with performance-, LLP-, and hierarchy-based goal values was somewhat difficult to distinguish from the model with performance- and LLP-based values alone.

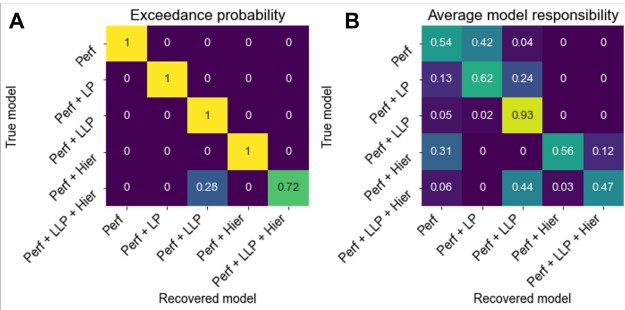

Figure 7: Confusion matrices for model recovery. **(A)** Exceedance probabilities for true and recovered models. **(B)** Average responsibilities for true and recovered models. Model name abbreviations: Perf = performance, LP = learning progress, LLP = latent learning progress, Hier = hierarchy.

# F   Options reinforcement learning model for simulations

In the experiment, participants made two kinds of choices, first picking a goal and then selecting a sequence of actions to try and achieve it. Since our focus was on the former, we only fit the data on the goal selection process, gathering information about action selection and feedback (required, e.g., for the computation of the performance, LP, and LLP features that entered the goal selection component) directly from the participants' data. However, simulating goal selection data – e.g., for parameter recovery (Figure 6), model recovery (Figure 7), and model validation (ensuring the model captures key aspects of the original data through simulations with the best-fit parameters [90, 91], Figure 2E) – requires creating a model that also generates data about the learning component. To this end, we developed models that attempted to learn the correct action sequences in addition to choosing their own goals. The goal selection component of the various models was described in Section 3.2. The learning component detailed below was shared across all models.

Each trial begins with a goal selection step (forced, in the training phase, or free, in the learning phase). Next, the agent decides which actions to pursue. The model's fundamental architecture is that of a standard $Q$-value reinforcement learning model [67], except that policies were goal-conditioned [70]. The model's $Q$-value table has shape $N_G \times N_A \times N_S$, where $N_G$ is the number of possible goals (6), $N_I$ is the number of ingredients available actions for selection at the current timestep, and $N_S$ is the total number of steps required to make the active goal (either two or four; Appendix B). Any of the four ingredients available for the active goal can be selected at the beginning of each trial and becomes unavailable once the agent chooses them. The model selects actions via a softmax policy [86] over the relevant $Q$-table until all the slots are filled:

$$P^t(a|g^*, h) = \frac{exp(\beta_{\text{learn}} \cdot F(Q^t(g^*, a)), h)}{\sum_i exp(\beta_{\text{learn}} \cdot F(Q^t(g^*, a_i)), h)}$$

where $t$ is the current trial, $a$ is the chosen action; $g^*$ is the active goal; $h$ is the current trial's history (i.e. the actions selected thus far) which, through the filter function $F$, determines the number of steps left to complete the trial and the actions the agent can consider taking next; $\beta_{\text{learn}}$ is the inverse temperature (the higher the $\beta_{\text{learn}}$, the more deterministic the decision).

Once all slots are full the agent receives a binary feedback $r^t$, equal to 1 if the chosen sequence of actions matched the active goal, and 0 otherwise. The agent then uses this signal to update the value

of each chosen action through the delta rule [87]. The filter function $F$ serves to address the correct $Q$ value based on the timestep at which the action to update was chosen:

$$F(Q^t(g^*, a), h) = F(Q^{t-1}(g^*, a), h) + \alpha_{\text{learn}} \cdot (r^t - F(Q^{t-1}(g^*, a), h))$$

where $\alpha_{\text{learn}}$ is the learning rate. This update function reflects the fact that the agent does not receive partial feedback for correct actions at the right timesteps, and can therefore only identify the appropriateness of chosen actions if the entire chosen sequence is correct.

The model's $Q$ table was also subject to forgetting as governed by the variable $phi$, which gradually brought back all $Q$ values to the initial value $Q_0$ (0):

$$Q^t(g, a) = Q^{t-1}(g, a) + \phi \cdot (Q_0(g, a) - Q^{t-1}(g, a)) \forall g \in G, a \in A$$

Except for the temporal aspect given by the nature of the environment, the model described until now is not capable of recognizing hierarchical relationships among goals. To enable it to leverage hierarchical structures and speed up learning, we endowed the model with the ability to build "options", i.e. to create and select sub-sequences of actions as a single chunk [92]. Upon receiving positive feedback, the model extends the space of possible actions with the chosen sequence of actions, which can then be considered a viable option whenever the ingredients it requires are available and the slots to fill are of at least the same length as the option itself. Options are selected and then updated in the same way atomic actions are. Note that, in reality, participants could also leverage insights from goals that shared structural similarity without requiring the same atomic actions (e.g., compare G3 and G5, Figure 1, bottom). While found that the simpler current model was sufficient to capture many aspects of the participants' behavior, future extensions could consider generalization rules that enable the model to share more abstract knowledge across goals.

The options model was unfittable through HBI methods due to intractable likelihood [101]. To approximate the learning parameters for each participant ($\beta_{learn}$, $\alpha_{learn}$, and $\phi$), we fit a simpler reinforcement learning model (without options), this time through maximum-likelihood methods and only on action selection date, and used the resulting best-fit learning parameters for simulations. To account for random noise, we ran 3 iterations per participant when simulating data for validation and model recovery.

## G    The role of hierarchy in learning and goal selection

Hierarchical features of the environment could interact with the goal selection process in at least two ways. First, hierarchy could have a direct role in goal selection, e.g., people might either prefer or avoid goals that enable them to exploit hierarchical relations. Second, hierarchical structures may impact individuals' learning, hence their LLP – which is then used to select goals. The fact that the winning model for goal selection had an LLP factor, but lacked direct information about the presence of hierarchical relationships among goals, suggests the latter hypothesis is more likely. However, we find that hierarchy plays a major role in the learning process, thereby affecting goal selection indirectly.

To illustrate this, we simulated participants' data with two models. The first model, used for all other simulations presented in this article, is a reinforcement learning with options ([92]; Appendix F), therefore able to exploit hierarchical relationships between simple and complex goals. The second model is a flat reinforcement learning model, which only has access to atomic actions. Despite the fact that both models lacked hierarchical factors in the goal selection stage, effects of hierarchy similar to those of real participants (Figure 2) were found in data simulated with the options reinforcement learning model (Figure 8A-C). By contrast, removing hierarchical information from the learning component completely removed hierarchy effects in learning and goal selection (Figure 8D-F). These results suggest at least some participants generated insights for learning by recognizing hierarchical relationships in the learning environment, while not necessarily using this information directly to decide which goals to select. Since the options model was not fittable through HBI methods, quantitative confirmation of these results awaits formal model comparison through more advanced model fitting methods [101].

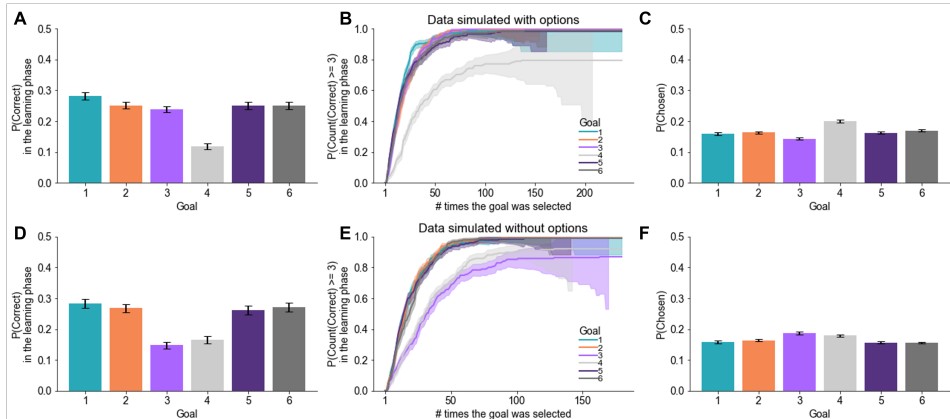

Figure 8: Simulations with two different learning models. **(A-C)** Simulated participants' behavior from a generating model with options shows hierarchy effects in learning and goal selection. **(D-F)** Simulated participants' behavior from a generating model without options fails to replicate hierarchy effects.

# H   Verbal reports of example participants' strategies

At the end of the experiment, participants were asked questions about the strategies they followed while playing the online game (Appendix B). Capital letters in the responses transcribed below correspond to participants in the subplots of Figure 3:

A. "Yes, I tried to find the easiest potion recipes firsts (two ingredients) and only focus on the recipes one at a time. When I found one, I would move to the next easiest. Eventually, I knew all of them but needed to engrain them into my memory. I then repeatedly moved from one recipe to the next practicing. Finally, I began to randomly select potions as a means of testing and improving my memory."

B. "I tried using permutation skills I had learned in a class I had taken on discrete math. This was especially helpful for the potions that required 4 ingredients, where I permuted the possible solutions into different sets depending on what the first 2 ingredients were. For example, I would choose a first and second ingredient, which left me with 2 possible combinations for the remaining 2 ingredients. If neither of those sequences worked, I would choose a different second ingredient and try the 2 possible sequences that could be made with this new combination of first and second ingredients. If that again didn't work, I would choose the last possible second ingredient. If I was still not successful in finding the correct sequence, I would try a different first ingredient and attempt this entire process all over again."

C. "I used the one that let me win each time."

D. "No I didn't use a strategy because I was confused on what to do in the beginning of the game."

# I   Clustering methods

The first step towards separating participants into clusters was engineering features from the raw data to obtain distinct behavioral signatures. The final subset of features was selected due to its enabling reliable cluster partitions and the low correlation across features (Figure 9A):

1. P(Correct) at learning: proportion of trials of the learning phase in which the chosen goal was achieved.

2. P(G in [G1, G2, G5, G6]): proportion of trials in which a 2-action goal was selected.

3. P(G in [G1, G2, G3, G5]): proportion of trials in which a goal involved in hierarchical relationships was chosen.

4. S(Goals): entropy of the chosen goals.

5. S(Actions): entropy of the chosen actions.

6. P(Correct) at testing: proportion of trials of the testing phase in which the proposed goal was achieved (including G7 and G8).

To capture temporal dynamics in participants' behavior, all features were computed separately for each of the six blocks of the learning section, except for P(Correct) at testing, which was a single value per subject since there was a single testing block. As a result, the behavioral features were highly dimensional. Thus, before performing clustering, we reduced the dimensionality of the features through principal component analysis (PCA). We identified 11 as the optimal number of principal components (PCs) through the elbow method over the cumulative explained variance. Plotting the PC loadings of the original features suggested PCA maintained the structure inherent in the raw features while enabling a more tractable dimensionality (Figure 9). For example, PC1 seemed to track performance indicators, PC2 defined action entropy early in learning, and PC3 represented goal entropy.

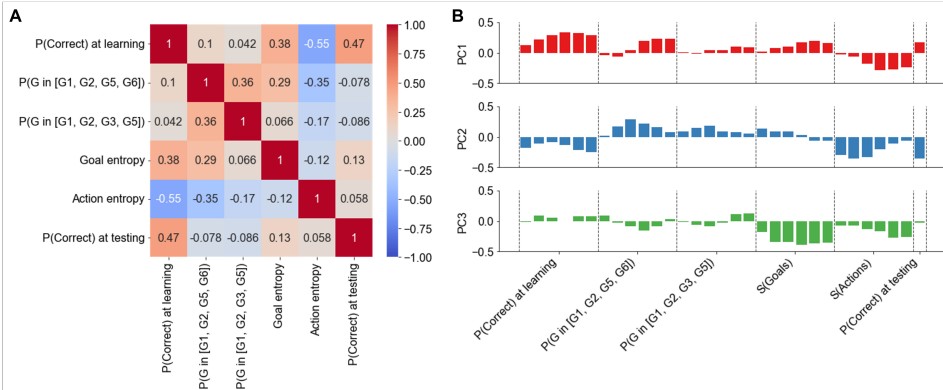

Figure 9: PCA feature correlations and loadings. (**A**) Pearson's correlations among features used for PCA. (**B**) Loadings of the chosen features on the first 3 PCs.

Participants were then clustered through the K-means method [93] over the PCs. However, similar clusters were obtained through different partitioning methods. The elbow-point heuristic over the within-cluster sum of squares (WCSS) was used to identify 4 as the optimal number of clusters (Figure 10A). Plotting combinations of the first three PCs confirmed the presence of four distinct groups 10B.

We used archetypal analysis – a technique which searches extremal points ("archetypes") in multidimensional data ([96], software by [102]) – to visualize extreme behaviors and exemplify participants' strategies in each cluster. The adjusted mutual information between K-means cluster identity and the closest archetype for each participant was 0.74, confirming much overlap between the two classification methods (Figure 10C).

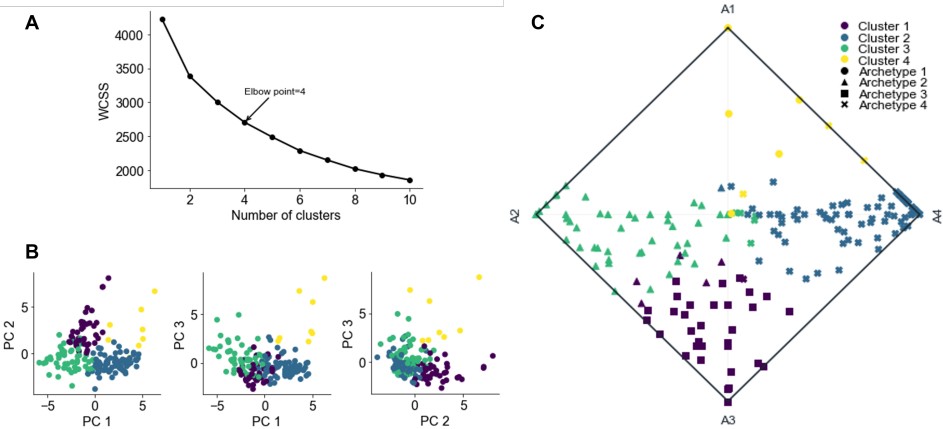

Figure 10: Clustering results. **(A)** Scree plot showing changes in WCSS as a function of the number of components used for K-means clustering. **(B)** Positioning of the identified clusters relative to the first three PCs. **(C)** Simplex plot showing archetype contributions to each participant, color-coded by the participant's K-means cluster identity and marker-coded by the archetype closest to each participant.

