# OpenReview forum: "Latent Learning Progress Drives Autonomous Goal Selection in Human Reinforcement Learning"
_NeurIPS.cc/2024/Conference — NeurIPS 2024 poster_

### Official Review · Reviewer_HMNe · 2024-06-24

**Soundness:** 4
**Presentation:** 3
**Contribution:** 3
**Rating:** 8
**Confidence:** 4

**Summary:**

This paper develops a new analysis of autonomous goal selection based on "latent learning progress". The key idea is that agents seek to maximize progress on a latent variable rather than an observed variable like performance. The paper reports an interesting experiment with humans that seeks to test whether people use latent learning progress to guide their goal selection.

**Strengths:**

- The core theoretcal idea (latent learning progress) is well-motivated and novel.

- The experiment is well-designed, interesting, and rigorously analyzed.

- The modeling approach is very thorough.

- The paper is clearly written.

- The literature review is comprehensive.

- Potentially impactful within cognitive science.

**Weaknesses:**

- Latent learning progress wasn't formally defined until well into the middle of the paper, and there it was only operationalized in terms of the specific experimental setup rather than something more general.

- While motivated by work in AI, the paper isn't really written in a way that will have a broad appeal to an AI audience. I'm not sure how much impact this work will have at least within AI.

Minor:

p. 1: "Along factors" -> "Along side factors"

p. 4: "in which factor" -> "in which factors"; "combination of factors" -> "combinations of factors"

**Questions:**

- Can the authors provide a more general formal definition of latent learning progress?

- How can latent learning progress be defined in such a way that it's likely to be both general and useful? My concern is that there are probably many versions of this which are not useful, depending on how one defines the latent variable whose progress is being monitored.

**Limitations:**

The authors have adequately described the paper's limitations. I don't see any potential negative societal impacts.

---

> ### Author Rebuttal · Authors · 2024-08-04
>
> We thank the reviewer for their positive assessment of our submission, and address their comments individually below.
>
> ---
>
> **Weaknesses**
>
> *Latent learning progress wasn't formally defined until well into the middle of the paper, and there it was only operationalized in terms of the specific experimental setup rather than something more general.*
>
> We provide an intuitive explanation of LLP in the introduction, and save a formal definition for later in the paper, where all other related concepts have been introduced, to facilitate the reader’s understanding of such definition. The point about a generalized version of the LLP definition was addressed in the global response.
>
> ---
> *While motivated by work in AI, the paper isn't really written in a way that will have a broad appeal to an AI audience. I'm not sure how much impact this work will have at least within AI.*
>
> As noted above, NeurIPS welcomes submissions from “Neuroscience and cognitive science” – therefore, we invite the reviewer to consider our contribution to such fields. That said, we provide several connections to the artificial intelligence and machine learning literature in the Introduction, Related Work, and Discussion sections of the article, and would welcome the reviewer’s suggestions to further bridge across different fields.
>
> ---
>
> **Minor**
>
> *p. 1: "Along factors" -> "Along side factors"*
> *p. 4: "in which factor" -> "in which factors"; "combination of factors" -> "combinations of factors"*
>
> Minor typos have been fixed in the camera-ready version.
>
> ---
>
> **Questions**
>
> *Can the authors provide a more general formal definition of latent learning progress?*
>
> This point was addressed in the global response.
>
> ---
>
> *How can latent learning progress be defined in such a way that it's likely to be both general and useful? My concern is that there are probably many versions of this which are not useful, depending on how one defines the latent variable whose progress is being monitored.*
>
> This point was addressed in the global response.
>
> ---
>
> **Limitations**
>
> *The authors have adequately described the paper's limitations. I don't see any potential negative societal impacts.*
>
> We included potential societal impacts in our initial submission, but we expanded it in response to reviewer y4se: “As our knowledge of human goal setting becomes more precise, however, ethical concerns regarding the use of behavioral sciences in marketing and management should be addressed, particularly in cases where highly personalized methods of influence could transform advertising into manipulation.”

---

> > ### Comment · Reviewer_HMNe · 2024-08-11
> > **response**
> >
> > Thank you for these responses, which address my comments. My score was already high (8), and I don't think the work justifies a higher score (9, which would imply it is "groundbreaking"), but I will advocate for its acceptance.

---

> > > ### Author Response · Authors · 2024-08-11
> > >
> > > We are glad to hear we were able to address all comments! Thank you again for the constructive feedback.

---

### Official Review · Reviewer_Grub · 2024-07-12

**Soundness:** 4
**Presentation:** 4
**Contribution:** 3
**Rating:** 7
**Confidence:** 4

**Summary:**

This paper examines how humans select between different possible goals. The authors designed a hierarchical reinforcement learning task where the main dependent variable was which goal participants chose to work on for each trial. Then, they built descriptive computational models to assess what parameters explained participants' goal choices throughout the duration of experiment. The parameters that were most predictive of the participants' goal selections were performance and latent learning progress.

The experimental task introduced in this paper bears a resemblance to Little Alchemy (https://littlealchemy.com/) insofar as participants combine together different elements to create new elements. Where the task differs is that, rather than freely combining elements without any specific goal in mind, the participants in this task had 6 (not including the testing phase) goal potions to create and they received feedback about whether or not they successful concocted their target potion for the trial.

Their findings contribute to understanding human learning and building autotelic machines. In terms of human learning, the paper provides new evidence that latent learning progress (rather than standard learning progress) drives people to chose some goals over others. In terms of building autotelic machines, new artificial agent models could incorporate latent learning progress when choosing goals to better mimic the adaptive and rapid learning of humans and animals.

**Strengths:**

Goal selection (and intrinsic motivation more broadly) is a fundamental question in the study of both biological and artificial intelligence. This paper tackles a significant research question by using a novel experimental paradigm. The original paradigm introduced in this paper could also be modified for future work to address other questions like open-ended learning or could be made more broadly available online to test more diverse pools of participants in the future.

The writing is very clear through out (and any lingering questions I had about the experimental task were clarified effectively in the appendix).

Another wonderful strength of the paper is their ability to report on inter-individual differences. They are able to fit descriptive models to each individual subject.

**Weaknesses:**

While this paper has a number of important strengths, the main weakness is that it is not clear how well these findings would generalize to different tasks (especially more real-world tasks):
- First, the experimental paradigm is very specific. Users have to pick from a limited list of goals with the explicit over-arching goal of being able to create all of the potions successfully. Conversely, people usually generate their own goals (e.g., "I'd like to read this book", "I'd like to go on cross-country trip to visit Aunt Ida") rather than picking from a specific list, and they often choose these goals without any supervised over-arching goal.
- Second, the participants were university students (mostly female) participating for course credit. Would the results generalize to different populations?
- Third, the measure of latent learning progress (LLP) is entirely specific to this experimental task. The authors operationally define latent learning progress as 1 - (N actions sequences tested / N possible action sequences). This measure only applies in situations where there are a finite number of options to try. How can future researchers extend this notion of LLP to open-ended tasks that use high-dimensional sensory and motor spaces (where this operational definition would not work anymore)?

To be clear, I don't expect the authors to design a new experimental paradigm or test a new demographic range of participants during the rebuttal period. But it would be helpful for them to address these as limitations.

On a separate note, the statistical analyses need effect sizes. (For example, a simple Cohen's d for each Wilcoxon / t-test would be suffice.) Some of these differences in task performance are statistically significant because p < .05, but are not necessarily behaviorally significant because the difference (effect size) is so small. Also, if I'm reading the stats correctly, the results in text do not match Fig 2A: Complex hierarchical G3 mean performance is reported as .37, but the bar in Fig 2A looks like it couldn't be any higher than .3.


Minor:
- The in-text references are sometimes formatted a little strangely in terms of the number (for example citing 12-17, 11 instead of citing 11-17 or citing 1, 33, 34, 3, 35, 5, 6, 36, 37, 8 instead of citing 1, 3, 5, 6, 8, 33-37).
- The explanation of the experimental game (section 3.1) is not that clear. The figure (Fig 1) is definitely helpful, and the description in the appendix clarifies the game really well. You might consider reformatting the paragraph describing all of the goals into a list format.
- Figure 3 is difficult to understand as-is, but I think it might actually be a really interesting figure. Part of what makes it unclear is the y-axis: What is the "action sequence index"? I suspect there might be an alternative to a dot plot that would be easier to read (maybe even forgoing the y-axis and showing a single strip of color-coded trials for each subject perhaps?). It would also help the readers to add some descriptors to each graph so the reader knows which elements to look for (e.g., setting a few objectives rather than trying all goals, unprincipled switching between goals).

**Questions:**

My questions are covered in the Weaknesses portion.

**Limitations:**

See Weaknesses section.

---

> ### Author Rebuttal · Authors · 2024-08-04
>
> We thank the reviewer for their positive comments and their extremely detailed, helpful feedback. We address their suggestions for improvement below.
>
> ---
>
> **Weaknesses**
>
> *First, the experimental paradigm is very specific. Users have to pick from a limited list of goals with the explicit over-arching goal of being able to create all of the potions successfully. Conversely, people usually generate their own goals (e.g., "I'd like to read this book", "I'd like to go on cross-country trip to visit Aunt Ida") rather than picking from a specific list, and they often choose these goals without any supervised over-arching goal.*
>
> We completely agree with the reviewer! In fact, we noted “In our task, people chose goals from a predefined menu of options. Although this facilitates the study of goal setting, people often invent their own goals by combining observations and imagination”. Having a highly controlled setup in which to study goal selection was essential for a first understanding of people’s approach, and our clustering analyses and plots of individual behaviors highlight vast variability in goal selection strategies even in such a limited space. Building on our initial findings, it may be possible to start exploring how people create and pursue more free-form goals (if not as rich as the examples mentioned by the reviewer, at least more self-designed than the ones we studied here). We note that our is a standard approach to studying a novel, complex question: we start from a simplified experimental design to establish findings in well controlled setting, and plan to adapt our approach to increasingly more naturalistic, complex settings to pursue a more nuanced understanding.
>
> ---
> *Second, the participants were university students (mostly female) participating for course credit. Would the results generalize to different populations?*
>
> This point was addressed in the global response.
>
> ---
> *Third, the measure of latent learning progress (LLP) is entirely specific to this experimental task. The authors operationally define latent learning progress as 1 - (N actions sequences tested / N possible action sequences). This measure only applies in situations where there are a finite number of options to try. How can future researchers extend this notion of LLP to open-ended tasks that use high-dimensional sensory and motor spaces (where this operational definition would not work anymore)?*
>
> This point was addressed in the global response.
>
> ---
>
> *On a separate note, the statistical analyses need effect sizes. (For example, a simple Cohen's d for each Wilcoxon / t-test would be suffice). [...].*
>
> We now provide the effect size for all Wilcoxon tests as the standardized effect size r (Z/√N; Rosenthal et al., 1994; see attached PDF, “Updated statistics with effect sizes”). Note that we have replaced the symbol “Z” with “W” for non-standardized test statistics in Wilcoxon tests.
>
> ---
> *Also, if I'm reading the stats correctly, the results in text do not match Fig 2A: Complex hierarchical G3 mean performance is reported as .37, but the bar in Fig 2A looks like it couldn't be any higher than .3.*
>
> We are grateful to the reviewer for noticing this mismatch! The stats are correct, and we have updated the figure to match them (see attached PDF, “Updated Figure 2”). The previous figure was also correct, but aggregated the data slightly differently. The two are now consistent.
>
> ---
>
> **Minor**
>
> *The in-text references are sometimes formatted a little strangely in terms of the number [...].*
>
> We thank the reviewer for noticing the citation formatting error, which we have fixed for the camera-ready version.
>
> ---
> *The explanation of the experimental game (section 3.1) is not that clear. The figure (Fig 1) is definitely helpful, and the description in the appendix clarifies the game really well. You might consider reformatting the paragraph describing all of the goals into a list format.*
>
> We thank the reviewer for their suggestion and for consulting the Appendix thoroughly. We had initially placed a full description of the task in the main body of the paper but had to move it to the Appendix due to space limitations. We intended to provide the reader with enough information about the task to understand the main takeaways and refer interested readers, with a more specific interest in cognitive science experiments, to details in the Appendix.
>
> ---
> *Figure 3 is difficult to understand as-is, but I think it might actually be a really interesting figure. Part of what makes it unclear is the y-axis: What is the "action sequence index"? I suspect there might be an alternative to a dot plot that would be easier to read (maybe even forgoing the y-axis and showing a single strip of color-coded trials for each subject perhaps?). It would also help the readers to add some descriptors to each graph so the reader knows which elements to look for (e.g., setting a few objectives rather than trying all goals, unprincipled switching between goals).*
>
> We thank the reviewer for their insightful analysis of Figure 3. “Action sequence index” refers to the unique combination of ingredients selected by the participants and their order (e.g., the first point would be [0, 1, 2, 3] – where 0 is the top-most ingredient on the screen, 1 the second, etc.). To improve clarity, we have relabeled the y-axis “Action sequence”, since the index is not defined in the text. We omitted the axis labels to avoid cluttering the figure, making it unnecessarily large, and overloading the reader with information. We used the preceding paragraph to guide the reader through specific aspects of the figure they should be focusing on, and would therefore find it redundant to repeat the information in the figure caption.

---

> > ### Comment · Reviewer_Grub · 2024-08-12
> >
> > Thank you for your thorough response. I've reviewed your general response and your responses to me and the other reviewers.
> >
> > I'm enthusiastic that your team plans "to adapt our approach to increasingly more naturalistic, complex settings to pursue a more nuanced understanding." Can you provide any more clarity about how the present study provides an entry point towards this goal? Once the settings are more naturalistic and complex, it seems like you won't be able to use the same computational definition of LLP, so you'll (computationally) need to start from "square one." Do you see the entry point as being the task, which could be adapted to be more open-ended? Or do you see the entry point as the overall theoretical idea of LLP?
> >
> > In terms of effect sizes, thank you for including these for the revision. For future studies, I encourage the authors to consider effect sizes that aren't normalized by the number of participants. When effect sizes are measurements that are normalized by sample size, it obscures the actual magnitude of the effect. (The raw magnitude/size of the effect is the same whether you test 50 or 500 people; the difference is that 500 will give you more statistical power.) This comes back to my original comment: "Some of these differences in task performance are statistically significant because p < .05, but are not necessarily behaviorally significant because the difference (effect size) is so small."

---

> > > ### Author Response · Authors · 2024-08-13
> > >
> > > We thank the reviewer for taking into consideration all our responses.
> > >
> > > We are grateful for their suggestion on effect sizes, which we will consider for future studies, although we decided to take a relatively standard approach here.
> > >
> > > As for entry points into more naturalistic studies, we view both paths the reviewer mentioned as interesting! We think our study provides an entry point into the study of goals in more naturalistic settings in at least three ways, including the two proposed by the reviewer:
> > > - [Feasibility] This study serves as an entry point in that it establishes the validity of LLP in a simple environment, preparing the ground for more complex situations.
> > > - [Task] The task we developed in this study could serve as a baseline for future work. One possible extension of the task would involve participants coming up with their own “goal potions” rather than selecting them from a predefined menu. In this case, LLP could still be defined in a similar way, provided that the number of possible combinations to achieve the participant-defined goal is known. Another possibility would be to use data from online videogames to study LLP, as was previously done in studies on motivation and (the standard version of) LP ([Brändle et al., 2024](https://osf.io/vg8dz)).
> > > - [Theory] The theoretical idea of LLP, could be computationally expanded to more complex internal models than simply the number of possible combinations.
> > >
> > > We hope this further clarifies the points raised by the reviewer.

---

> > > > ### Comment · Reviewer_Grub · 2024-08-13
> > > >
> > > > Thanks for your response. I'm especially looking forward to seeing how the authors will computationally expand on LLP in the future. (And I appreciate that the authors likely do not want to go into too much detail on how they plan to do so quite yet.) In the meantime, I hope that we'll be seeing this paper at NeurIPS, and I'm raising my score accordingly to reflect that.

---

> > > > > ### Author Response · Authors · 2024-08-13
> > > > >
> > > > > Thank you for your careful feedback, enthusiasm, and support!

---

### Official Review · Reviewer_y4se · 2024-07-13

**Soundness:** 4
**Presentation:** 4
**Contribution:** 4
**Rating:** 8
**Confidence:** 3

**Summary:**

This paper looks at human goal selection during learning. A known useful signal for goal selection is learning progress (LP). LP measures performance from past observations, and is therefore only sensitive to measurable change in performance. The paper hypothesizes that goal selection is additionally driven by a “latent LP” (LLP) measure, which an agent infers from its knowledge of and interaction with the environment, and which therefore does not depend solely on observable performance improvements. Computationally, for each potential goal that can be set, LP is modeled as a value that is updated based on changes in the performance prediction error, while LLP is modeled as tracking how much of the solution space is currently unexplored (thus, it is a proxy for how close the agent is to solving the goal). A simple goal selection model is proposed under which goals are selected with probability given by a softmax function of goal-specific values. These values are a weighted sum of - among other factors - the aforementioned LP and LLP values. An experimental setup is then introduced in which participants were asked to iteratively select one of many goals and to attempt to solve that goal. The fact that participants had to actively select goals made participants’ goal selection behavior directly observable. Data from human participants reveals that the LLP factor is an important element in goal setting behavior.

**Strengths:**

This is a very good paper. The writing is clear and the motivation and discussion of prior work in the introduction and section 2 is excellent. Understanding human goal selection is important. Not just in its own right, but also for human-AI interaction settings where better models of human behaviour can help to train better AI assistants or companions. Furthermore, a better understanding of how humans direct their learning may inspire future research on things like curriculum design for RL. The experiments presented in this paper are novel, rigorous, and clearly support the claim that LLP informs goal selection. Moreover, there is a wealth of additional information and additional results in the appendices.

**Weaknesses:**

The definition of LLP introduced here is quite specific to a small and discrete space of action sequences. A more general definition or some preliminary discussion on how one could extend the current work to continuous or countably infinite spaces of action sequences would strengthen the work.

**Questions:**

The goal selection model considers a weighted sum for combining the factors that contribute to the goal values. Did you consider or test more complex functions of the factors?

**Limitations:**

Limitations are discussed in the conclusion. The authors do mention that potential future ethical concerns should be addressed, but do not expand on what those concerns may be.

---

> ### Author Rebuttal · Authors · 2024-08-04
>
> We thank the reviewer for their excellent summary of our article and their thorough feedback. We address their suggestions for improvement below.
>
> ---
>
> **Weaknesses**
>
> *The definition of LLP introduced here is quite specific to a small and discrete space of action sequences. A more general definition or some preliminary discussion on how one could extend the current work to continuous or countably infinite spaces of action sequences would strengthen the work.*
>
> This point was addressed in the global response.
>
> ---
>
> **Questions**
>
> *The goal selection model considers a weighted sum for combining the factors that contribute to the goal values. Did you consider or test more complex functions of the factors?*
>
> In our initial analyses, we indeed considered more complex functions of the factors. . Specifically, we tried modeling more complex interactions in our current setup (e.g., interactions between performance and other goal selection motives), but did not find this procedure to improve fit. Moreover, we note that we already observe interesting dynamics with a simple model, which is preferable for interpretability. It is possible that more complex functions of the factors might be relevant in more complex settings, and this will be an important question for future research. Here, we follow the standard “Occam’s razor” approach of the lowest sufficient complexity of the model given our data (Wilson and Collins, 2019).
>
> ---
>
> **Limitations**
>
> *Limitations are discussed in the conclusion. The authors do mention that potential future ethical concerns should be addressed, but do not expand on what those concerns may be.*
>
> We thank the reviewer for the opportunity to expand on such concerns, and we now state that : “As our knowledge of human goal setting becomes more precise, however, ethical concerns regarding the use of behavioral sciences in marketing and management should be addressed, particularly in cases where highly personalized methods of influence could transform advertising into manipulation.”

---

> > ### Comment · Reviewer_y4se · 2024-08-11
> >
> > Thank you for the clarification, and for addressing the potential ethical concerns. I agree entirely with your justification for using a weighted sum of factors.
> >
> > I think this is a really strong paper, and agree with the authors' view that it is suitable for NeurIPS. Based on the authors' responses to my review and the other reviews, I have decided to raise my score.

---

> > > ### Author Response · Authors · 2024-08-11
> > >
> > > We are happy to hear that all points were clarified. Thank you for your constructive feedback and for supporting our contribution to NeurIPS!

---

### Official Review · Reviewer_kdQC · 2024-07-13

**Soundness:** 3
**Presentation:** 2
**Contribution:** 2
**Rating:** 6
**Confidence:** 3

**Summary:**

This work presents a hypothesis of a latent learning process that can guide autotelic agents in goal selection. Human experiments provide evidence supporting this hypothesis.

**Strengths:**

Autotelic agents represent an important research direction. This work provides evidence from human experiments on latent learning processes, which could later be used to develop effective and personalized learning progressions for human-like autotelic machines.

**Weaknesses:**

1. All the subjects of the human experiment are students, which may introduce bias.
2. The definition of $V_f$ for each kind of model is unclear: e.g. in "performance", what does "currently active goal" mean? Where does $r^t$ come from? What is the relationship between it and the goal?
3. The design of LLP, which involves selecting multiple action sequences, does not provide a complete explanation of how LLP can improve human choices or actions better than LP.

**Questions:**

1. Hierarchy influences goal selection through latent learning processes (LLP). Will separating the evaluation of hierarchy and LLP affect the results?
2. Section 6 discusses individual differences. Has there been an analysis of the causes behind these differences?

**Limitations:**

Yes.

---

> ### Author Rebuttal · Authors · 2024-08-04
>
> We thank reviewer kdQC for examining our submission and recognizing its importance and potential impact on machine learning. In the rebuttal period, we hope the reviewer could also help clarify what would improve soundness and presentation beyond the point raised so far. Regarding contribution, we would like to stress the fact that NeurIPS welcomes submissions from “Neuroscience and cognitive science” – therefore, we invite the reviewer to consider our contribution to such fields. We address all of the reviewer’s stated concerns below.
>
> ---
>
> **Weaknesses**
>
> *All the subjects of the human experiment are students, which may introduce bias.*
>
> This point was addressed in the global response.
>
> ---
> *The definition of  V_f for each kind of model is unclear: e.g. in "performance", what does "currently active goal" mean? Where does r^t  come from? What is the relationship between it and the goal?*
>
> The currently active goal is the goal a participant has chosen on a given trial. To make our statement clearer, we now say “selected goal” instead of “currently active goal”. $r^{t}$ comes from the feedback provided to participants (the target potion either fills up or remains empty), as specified in the same sentence:  “1 for positive feedback, 0 for negative feedback”. Feedback is goal-contingent, such that participants only receive feedback relative to the selected goal. We clarify this further in the text: “On each trial, the utility of the selected goal with respect to performance is updated based on the goal-contingent feedback received on that trial $r^{t}$”.
>
> ---
> *The design of LLP, which involves selecting multiple action sequences, does not provide a complete explanation of how LLP can improve human choices or actions better than LP.*
>
> As mentioned in the paper, “unlike LP, LLP does not require observing performance improvements to provide informative signals about progress.” This would make LLP particularly useful in situations (mentioned in the paper) “where no external change is visible, yet some other form of progress toward the desired outcome is made. Imagine being tasked with identifying the correct sequence of numbers to open a combination lock, which you might attempt through trial and error. Throughout most of this scenario, repeated failures would yield no difference in performance, hence no empirical learning progress (as typically defined). Nonetheless, provided that the lock has a limited number of slots and numbers and that you can avoid repeating incorrect combinations, every attempt is a step toward the solution.”
>
> ---
> **Questions**
>
> *Hierarchy influences goal selection through latent learning processes (LLP). Will separating the evaluation of hierarchy and LLP affect the results?*
>
> Hierarchical components are a design choice in our experimental paradigm. Removing the hierarchical feature would likely make it more difficult to tease LLP and LP apart. We find, however, that exactly how hierarchy and LLP relate to and influence each other is an interesting question. Our current findings suggest that “hierarchy impacts goal selection indirectly by enabling inferences and thus affecting LLP”.
>
> ---
>
> *Section 6 discusses individual differences. Has there been an analysis of the causes behind these differences?*
>
> The goal of the current paper was not to identify the sources of individual differences in goal selection. Noting that such differences exist is in fact a novelty of our submission. That said, we agree with the reviewer that the question of individual differences is worthy of further exploration, and we mention it would be interesting if future studies crossed findings similar to ours with participants’ data related to “demographics and cultural background, cognitive abilities, and psychopathology”.

---

> > ### Comment · Reviewer_kdQC · 2024-08-10
> >
> > Thank you for the response, which addressed most of my questions.
> > The only unclear thing left is about the model fitting described in lines 208-215. Could you clarify what the fitted parameters and loss function are in this fitting process? How did you determine the 'responsibilities' of each model in explaining participants' behavior?

---

> > > ### Author Response · Authors · 2024-08-10
> > >
> > > We are glad that the reviewer found our response exhaustive, and thank them for bringing up an additional question. Due to space limitations, we did not fully expand on Hierarchical Bayesian Inference (HBI) in the manuscript and instead redirected the reader to the original article that introduces this model fitting method ([Piray et al., 2019](https://doi.org/10.1371/journal.pcbi.1007043)). The cited paper contains full details and information, as well as a [GitHub repository](https://payampiray.github.io/cbm) with the code to implement it.
> > >
> > > Briefly, HBI performs concurrent model fitting and model comparison. It characterizes a population-level distribution of parameters from which individual estimates are drawn, in a way that is proportional to the probability of each subject’s data being generated by each model i.e., the model’s responsibility with respect to each subject. Large values of responsibility (close to 1) for a subject and model indicate the model is likely to be the best underlying model for the subject. The HBI algorithm comprises four steps, which are iterated on until stopping criteria are met: 1) calculate the summary statistics, 2) update posterior estimates over group parameters, 3) update the posterior estimate over each individual parameter, 4) update estimates of each model’s responsibility with respect to each individual subject. For individual parameters, we used the default priors of 0 for the mean and 6.25 for the variance. We refer the reviewer to the original article ([Piray et al., 2019](https://doi.org/10.1371/journal.pcbi.1007043)) for all mathematical details.
> > >
> > > While relatively new, this method has been used successfully in several applications (at the time of writing, Piray et al., 2019 counts over 100 citations). HBI outperforms more traditional statistical tools, such as maximum likelihood model fitting, as it is less prone to overfitting and less likely to favor overly simplistic models (see [Piray et al., 2019](https://doi.org/10.1371/journal.pcbi.1007043)).
> > >
> > > We hope this answer satisfies the reviewer. Given that all other points were addressed in our previous response, we hope the reviewer will consider increasing their score, or help us address any remaining questions by stating them in a comment.

---

> > > > ### Comment · Reviewer_kdQC · 2024-08-11
> > > >
> > > > The new response addressed my question. I would like to increase the score.

---

> > > > > ### Author Response · Authors · 2024-08-11
> > > > >
> > > > > We are glad to hear all questions have been addressed and see the score has been increased. Thanks again for your constructive feedback!

---

### Author Rebuttal · Authors · 2024-08-04

We would like to express our gratitude for the reviewers’ positive and constructive feedback.

The reviewers highlighted several strengths of the paper, including the important research direction it falls in line with and its tackling a fundamental question. They noted the potential impact of our findings on research in cognitive science, but also the design of autotelic machines and the development of AI assistants or companions that better align with human goals. The reviewers also appreciated the novelty of both the theoretical idea and the empirical approach, which could be extended to further address questions on open-ended learning. According to the reviewers, the writing was clear, the supported evidence was comprehensive, and the analysis thorough.

The reviewers also noted some limitations. We address some common questions below, and invite reviewers and ACs to refer to individual responses for other points raised in the review process. We hope the provided answers and clarifications help improve our submission’s score wherever possible, and otherwise kindly ask the reviewers to express remaining concerns so we can best address them.

Reviewers kdQC selected “fair” as a contribution score, and HMNe mentioned that the paper’s appeal to an AI audience could be improved. First, we would like to point out that NeurIPS welcomes (and has welcomed in the past) submissions from “Neuroscience and cognitive science” (as mentioned in the [call for submissions](https://neurips.cc/Conferences/2024/CallForPapers)), where we find our paper makes the most immediate contributions. That said, we made sure to provide several connections to the artificial intelligence and machine learning literature in the Introduction, Related Work, and Discussion sections of the article. Reviewer Grub perfectly captured our positioning in terms of impact, stating that “In terms of human learning, the paper provides new evidence that latent learning progress (rather than standard learning progress) drives people to choose some goals over others. In terms of building autotelic machines, new artificial agent models could incorporate latent learning progress when choosing goals to better mimic the adaptive and rapid learning of humans and animals”.

Reviewers kdQC and Grub pointed out that our participants were students from a university cohort. We would like to note that this is standard practice in the field. We, as well as other researchers studying higher cognition in various domains, have observed that qualitative effects found in university student populations typically generalize well to a broader, more diverse population of healthy adults. The reviewers are, however, correct that we cannot guarantee that our findings will indeed generalize to the general population and that this will be more important as we consider individual differences alongside whole-group qualitative effects. We recognized this limitation in our initial submission: “However, our sample was restricted to a relatively homogeneous group of undergraduate university students.” We now also mention that “Future studies and computational models may extend participation to a broader subject pool and specifically address individual differences in goal selection and achievement [...]”. Nonetheless, LLP was a key factor across participant clusters. While the distribution of clusters might change across populations (i.e., different strategies might be more or less common in certain populations), we predict LLP to be widely used.

All reviewers noted the lack of a generalized definition of LLP beyond the scope of our experimental paradigm, especially with respect to situations where possible solutions are not finite. First, we note that while simplistic, our novel experimental paradigm is already quite complex relative to the existing setups. However, we agree with the reviewers this is a limitation of the work as it currently stands. As noted in the article, “While we provide a simple and task-specific formalization of LLP, a generalized definition is necessary to understand the differences between LLP and other intrinsic motivation signals and ease the implementation of LLP-based goal selection in autotelic machines”. However, our focus in this initial work was to provide evidence for LLP use in humans, and it was essential to establish this aspect to “inspire the establishment of even more precise signals for goal selection in both humans and artificial agents”. We also note that even in situations where infinite spaces are theoretically possible, humans often restrict possibilities to a countable amount, and can therefore keep learning by exclusion like participants in our study did. For example, in a search problem such as looking for one’s glasses, one could consider virtually infinite locations, but rarely does so – making our current definition of LLP easily applicable. In more complex, real-world situations, we speculate that people may rely on a different internal model of their environment and their non-observable progress as they attempt to solve it – however, future work will be needed to extend LLP in these situations and describe such internal model in full detail.

Overall, we find the reviewer’s feedback encouraging and constructive, and we look forward to continuing the discussion.

---

### Comment · Area_Chair_QzTD · 2024-08-12
**Discussion period almost over**

The discussion period is almost over, so both authors and reviewers please respond to any unaddressed questions. Reviewers, be sure that you have all of the information you need from the authors, since after the 13th, they won't be able to respond.

---

### Decision · Program_Chairs · 2024-09-25

**Decision:**

Accept (poster)

**Comment:**

While their were some initial concerns, the authors addressed everything that could be considered significant. Several reviewers consider this a particularly strong contribution in this area and I'm inclined to agree.